# Vitamin A Status in Preterm Infants Is Associated with Inflammation and Dexamethasone Exposure

**DOI:** 10.3390/nu15020441

**Published:** 2023-01-14

**Authors:** Madelaine Eloranta Rossholt, Kristina Wendel, Marianne Bratlie, Marlen Fossan Aas, Gunnthorunn Gunnarsdottir, Drude Fugelseth, Are Hugo Pripp, Magnus Domellöf, Ketil Størdal, Tom Stiris, Sissel Jennifer Moltu

**Affiliations:** 1Department of Neonatal Intensive Care, Oslo University Hospital, 0450 Oslo, Norway; 2Department of Pediatrics and Adolescence Medicine, Oslo University Hospital, 0450 Oslo, Norway; 3Institute of Clinical Medicine, University of Oslo, 0450 Oslo, Norway; 4Department of Pediatric Neurology, Oslo University Hospital, 0450 Oslo, Norway; 5Oslo Centre of Biostatistics and Epidemiology, Oslo University Hospital, 0450 Oslo, Norway; 6Department of Clinical Sciences, Pediatrics, Umeå University, 901 87 Umea, Sweden

**Keywords:** vitamin A, preterm infant, dexamethasone, inflammation, bronchopulmonary dysplasia

## Abstract

Vitamin A has a key role in lung development and its deficiency is associated with an increased risk of bronchopulmonary dysplasia. This secondary cohort analysis of the ImNuT trial (Immature, Nutrition Therapy NCT03555019) aimed to (1) explore vitamin A status in preterm infants <29 weeks gestation and (2) assess the influence of inflammation and postnatal dexamethasone exposure on vitamin A concentrations in blood. We report detailed information on vitamin A biochemistry, vitamin A intake, markers of inflammation and dexamethasone exposure. After four weeks of age, infants exposed to dexamethasone (*n* = 39) showed higher vitamin A concentrations compared to unexposed infants (*n* = 41); median (IQR) retinol was 1.0 (0.74, 1.5) vs. 0.56 (0.41, 0.74) µmol/L, *p* < 0.001. Pretreatment retinol concentrations were lower in the dexamethasone group compared to non-exposed infants (*p* < 0.001); 88% vs. 60% of the infants were considered deficient in vitamin A (retinol < 0.7 µmol/L) at one week of age. Small size for gestational age, mechanical ventilation and elevated levels of interleukin-6 were factors negatively associated with first-week retinol concentrations. In conclusion, preterm infants <29 weeks gestation are at risk of vitamin A deficiency despite intakes that accommodate current recommendations. The presence of inflammation and dexamethasone exposure should be considered when interpreting vitamin A status.

## 1. Introduction

Very preterm infants (born < 32 weeks gestation) are at high risk of concomitant diseases due to immaturity. Bronchopulmonary dysplasia (BPD) is a major morbidity affecting approximately 50% of survivors born <28 weeks gestation [1,2]. The pathogenesis is multifactorial, but arrested lung development and inflammatory processes seem to foster disease progression [3]. Mechanical ventilation and oxidative stress induce damage to the immature lungs and aggravate inflammation [3].

Vitamin A is a fat-soluble vitamin that refers to a group of compounds, including retinol, retinal and retinoic acid, which all play a range of functional roles [4]. It is regarded as an anti-inflammatory vitamin and is essential for immune competence [5]. Retinoic acid, a potent regulator of gene expression, modulates the transcription of genes that are critical in antioxidant responses [6]. Vitamin A is also crucial in key developmental processes in the lung; it is essential for the normal growth and differentiation of the respiratory epithelium [7] and it regulates the formation of alveoli and surfactant synthesis [8,9]. Very preterm infants are born with low vitamin A stores [10], and postnatal deficiency is associated with an increased risk of BPD [9,11,12,13,14]. Early enhanced supplementation with vitamin A improves respiratory outcomes in very preterm infants, but the beneficial effects might be limited to those with a baseline vitamin A intake below recommendations [15].

The liver is the main site of vitamin A storage, but some vitamin A is also stored in other tissues, including the lungs [7,16]. Measures of hepatic reserves are the “gold standard” for assessing vitamin A status [17], but estimations of body stores are not feasible in clinical practice. Therefore, blood concentrations of retinol are the most commonly used biomarker [17]. Retinol circulates in blood bound to retinol-binding protein 4 (RBP4), a negative acute-phase protein synthesized in the liver. In children and adults, the presence of a systemic inflammation response affects vitamin A homeostasis; the release of pro-inflammatory cytokines suppresses the synthesis of RBP4, leading to lower vitamin A concentrations in blood [17,18,19].

Administration of systemic corticosteroids (e.g., dexamethasone) after the first week of life reduces the risk of BPD through anti-inflammatory properties, particularly in infants who cannot be weaned from mechanical ventilation [20]. It has also been postulated that the therapeutic response to dexamethasone may partly be linked to the mobilization of endogenous vitamin A reserves during exposure, leading to the enhanced supply and utilization of vitamin A in the premature lungs [7,21,22,23,24].

The aim of the present study was to assess vitamin A status during hospitalization in preterm infants <29 weeks gestation. We hypothesized that blood concentrations of vitamin A differed among infants who received treatment with dexamethasone compared to those who did not. Secondly, we hypothesized that markers of inflammation were negatively associated with vitamin A status, and that these observations were evident in infants later diagnosed with BPD.

## 2. Materials and Methods

### 2.1. Study Design and Subjects

The present observational study was based on the cohort of preterm infants (<29 weeks gestation) who participated in the double-blind, randomized, controlled ImNuT trial (NCT03555019) conducted at Oslo University Hospital, Norway [25]. We report a secondary analysis on vitamin A status from inclusion to 36 weeks postmenstrual age (PMA). Study participation required written informed, parental consent within 48 h after birth. Exclusion criteria were major congenital malformations, chromosomal abnormalities and critical illness with short life expectancy. Study infants were randomized to receive 0.4 mL/kg of a daily enteral fatty acid supplement consisting of either medium-chain triglycerides (MCT-oil™, Nutricia, Utrecht, The Netherlands) or 100 mg/kg of arachidonic acid and 50 mg/kg of docosahexaenoic acid (Formulaid™, DSM Nutritional Products Inc., Glenville, NY, USA). None of the supplements contained vitamin A. The supplements were started within 48 h after birth, and provided until 36 weeks PMA. Details about the method of randomization and blinding are described in the study protocol [25].

### 2.2. Nutritional Management

The nutritional management of the ImNuT cohort followed a standardized feeding protocol to accommodate European guidelines [10,26]. The protocol has been published elsewhere [27]. Briefly, all infants received a combination of parenteral nutrition (PN) and human milk within a few hours after birth. Multicomponent fortification of human milk was initiated and advanced before PN was discontinued so that nutrient intakes were maintained throughout the transition to exclusive enteral feeds. All infants received a standard dose of an enteral fat-soluble multivitamin supplement including preformed vitamin A (retinyl palmitate). The supplement was initiated when enteral feeds were established, and thereafter administered once daily.

### 2.3. Data Collection and Nutritional Assessment

Nutrient intake: We prospectively assessed all enteral and intravenous intakes from birth to 36 weeks PMA using a computer-aided nutrition calculation program (Nutrium software by Nutrium AB, Umeå, Sweden). In infants transferred to their local hospitals before 36 weeks PMA, complementary nutrient data were collected from the local hospital. The day of birth was included in analyses as Day 0. Vitamin A intake was reported as microgram (μg) retinol per kg body weight per day (1 μg = 3.33 IU retinol). Birth weight was used for calculations of nutrient intake until birth weight was regained; otherwise, current weights were used for all calculations.

Biochemistry: Retinol (μmol/L) and RBP4 (μg/mL) were measured in 10 μL dried whole blood spot (DBS) samples obtained at inclusion (within 48 h of life), Day 7, Day 28 and Week 36 PMA. The samples were collected on Mitra^®^ sticks, dried in room air and stored at –80 °C until assayed. For this study, interleukin (IL)-6 and C-reactive protein (CRP) obtained at the same time points were included as markers of inflammation. The methods for retinol, RBP4 and IL-6 analyses are described in Appendix B. Plasma retinol was routinely measured at Day 28 and at 36 weeks PMA. Vitamin A deficiency was defined as retinol concentrations <0.7 µmol/L, and retinol <0.35 µmol/L was defined as severe deficiency [28].

Clinical outcomes: Neonatal morbidity data were prospectively registered in the ImNuT trial [25]. In our department, very preterm infants are considered for dexamethasone treatment according to the DART protocol [29] if they receive mechanical ventilation after 10 days of life. This treatment regime results in a cumulative dexamethasone dose of 0.9 mg/kg. BPD was defined as a need for respiratory support or oxygen at 36 weeks PMA [1], and the severity of BPD was classified into (1) mild or moderate and (2) severe disease [1].

### 2.4. Data Analyses and Statistics

Continuous variables are reported as the median and interquartile range (IQR) or range, and categorical variables as a proportion and percentage. We performed the Mann–Whitney U test for continuous variables and the χ^2^ test for categorical variables to compare vitamin A biochemistry and intake between groups. Fisher’s exact test was used for categorical variables if the group sample was <6. For the analysis of longitudinal measures of retinol and RBP4 concentrations, we used mixed models for repeated measurements with a subject-specific random intercept. For comparisons of the randomized groups, inclusion was used as a baseline and fixed effects in the models were the outcome measurements at baseline, treatment group, follow-up time (i.e., Day 7, Day 28 and 36 weeks PMA) and the interaction between treatment group and follow-up time. For the other comparisons, clinical group, time (i.e., inclusion, Day 7, Day 28 and 36 weeks PMA) and the interaction between clinical group and time were used as fixed effects. Infants who received dexamethasone treatment during hospitalization were referred to as the DEXA group. Infants diagnosed with BPD at 36 weeks PMA were included in the BPD group.

We explored graphs and correlations between vitamin A biochemistry and intake, as well as for possible confounders during the study period. Non-normally distributed variables were transformed on a log10 scale to correct for skewness. A backward stepwise linear regression model was carried out to explore the influence of possible predictors of retinol concentrations on postnatal Day 7. Sex, gestational age (GA), small for gestational age (SGA) status, Clinical Risk Index for Babies (CRIB) score 1, mechanical ventilation at Day 7, total vitamin A intake from birth to Day 7 and blood concentrations of CRP and IL-6 on a log10 scale were included as predictor variables hypothesized to be associated with vitamin A status. A *p*-value threshold of 0.2 was set as the elimination criterion for variables included in the final model. Missing values were excluded listwise. *p*-value < 0.05 denoted statistical significance. Analyses were performed using IBM SPSS Statistics version 26 and STATA statistical software Version 17.0.

## 3. Results

### 3.1. Study Population

Overall, 121 infants were included in the ImNuT trial between April 2018 and January 2021 (one exclusion after randomization). Of these, 110 infants were eligible for neonatal outcome and nutrition analyses at 36 weeks PMA (flow chart, Figure 1).

### 3.2. Vitamin A Concentrations

There were strong correlations between whole blood retinol and RBP4 concentrations at each assessment (r = 0.82–89, *p* < 0.001), and both correlated with plasma retinol concentrations: r = 0.83 and r = 0.89 (both *p* < 0.001). We therefore present our results on vitamin A status by referring to retinol concentrations (obtained from whole blood spot samples), if not otherwise stated. Vitamin A biochemistry did not differ between the intervention and control groups in the ImNuT trial (Appendix A). Hence, we present our data pooled for the ImNuT cohort, as well as split according to whether the infants received treatment with dexamethasone or not. Characteristics and clinical outcome data are summarized and compared between the groups in Table 1.

Z scores were derived from Niklasson 2008 [30]. Data are missing for patients who died or were withdrawn before 36 weeks PMA (^a^
*n* = 110) or before the first ROP screening (^b^
*n* = 110)—nine infants in the No DEXA and one infant in the DEXA group. Data are missing for patients who died or were withdrawn before the first echocardiography (^c^
*n* = 115)—five infants in the No DEXA group.

### 3.3. Vitamin A Deficiency

Vitamin A deficiency (retinol concentrations < 0.7 µmol/L) was evident in the majority of infants through hospitalization (Appendix A). Overall, 75% had retinol concentrations <0.7 µmol/L at inclusion. At 36 weeks PMA, 4 out of 5 infants had retinol concentrations <0.7 µmol/L, and 25% of the cohort showed severe vitamin A deficiency (retinol < 0.35 µmol/L).

### 3.4. Associations between Vitamin A and Dexamethasone Exposure

All 49 infants in the DEXA group received dexamethasone to facilitate extubation. All infants were older than one week of age at treatment initiation; median age was 17 (IQR 15, 20) days, and the duration of treatment was median 10 (IQR 9, 13) days. Changes in retinol concentrations showed a different pattern among infants in the DEXA group compared to those who were not exposed to dexamethasone (Figure 2). At Day 7, pretreatment retinol concentrations were lower in the DEXA group compared to non-exposed infants: median (IQR) 0.42 (0.30, 0.59) vs. 0.66 (0.51, 0.86) µmol/L, *p* < 0.001. From Day 7 to 28, there was a two-fold increase in retinol concentrations in the DEXA group (Figure 2).

The highest concentrations of retinol and RBP4 were observed during, or shortly after, dexamethasone exposure (Figure 3a,b). There was no difference in vitamin A biochemistry at inclusion or at 36 weeks PMA. A detailed between-group comparison on vitamin A biochemistry and intake is provided in Appendix A.

### 3.5. Associations between Vitamin A and Markers of Inflammation

Measurements obtained at inclusion and Day 7 showed a negative correlation between IL-6 and retinol concentrations (Figure 4). IL-6 levels were higher among infants in the DEXA group compared to the No DEXA group: median (IQR) 16 (10, 28) vs. 7 (4, 15), *p* < 0.001. CRP levels did not differ between the dexamethasone groups at any time point (data not shown). The median CRP level in the first week was 0.8 (IQR 0.6, 3.2) mg/L and did not correlate significantly with retinol concentrations: r_s_ = −0.138, *p* = 0.068 (Appendix A). Retinol concentrations did not correlate with markers of inflammation at Day 28 or at 36 weeks PMA.

### 3.6. Associations between Vitamin A Biochemistry and Intake

Provision of vitamin A was below recommendations during the first postnatal week, but increased with higher enteral intakes and with the initiation of vitamin supplementation (Appendix A). Overall intake from birth to 36 weeks PMA was within the recommended range in all participants.

First-week vitamin A intake was lower in the DEXA group compared to the No DEXA group (*p* < 0.001) (Appendix A). Total and enteral vitamin A intakes (but not parenteral intakes) were associated with retinol concentrations at Day 7: r_s_ = 0.256 (*p* = 0.012) and r_s_ = 0.296 (*p* = 0.003). We did not find any correlations between retinol concentrations and vitamin A intake at Day 28 and 36 weeks PMA (data not shown).

### 3.7. Multivariable Analyses between Vitamin A and Potential Predictors at Day 7

Mechanical ventilation at Day 7, being born SGA and higher IL-6 values were negatively associated with retinol concentrations (all *p* < 0.05). The adjusted R^2^ suggests that the multivariable linear regression model accounted for 37% of the variability in retinol concentrations at Day 7 (Table 2).

## 4. Discussion

In this cohort of preterm infants, the majority exhibited vitamin A deficiency (retinol < 0.7 µmol/L) despite vitamin A intakes within the upper range of recommendations after the first week of life [31]. We observed a two-fold transient increase in retinol concentrations during dexamethasone exposure. First-week retinol concentrations were lower in infants later diagnosed with BPD compared to those without BPD. Mechanical ventilation, being born SGA and higher levels of IL-6 were independently associated with poorer vitamin A status at one week of age.

The high occurrence of vitamin A deficiency at study inclusion was not surprising. Placental transport of vitamin A occurs predominantly during the third trimester of pregnancy, leading to lower concentrations in cord blood and serum in preterm infants compared to term-born peers [32]. In contrast to the No DEXA group, retinol concentrations dropped during the first postnatal week in the DEXA group. Infants later diagnosed with severe BPD exhibited the lowest concentrations at Day 7 (all, retinol < 0.7 µmol/L). These observations support previous studies suggesting that vitamin A deficiency may contribute to the development of BPD [13,14,33]. In 115 very preterm infants, first-week retinol concentrations <0.7 µmol/L were associated with eight-times higher odds of BPD compared to those with retinol >0.7 µmol/L after adjusting for GA, birth weight, patent ductus arteriosus (PDA) and pulmonary infection [33]. An association between vitamin A deficiency and BPD independent of GA has also been reported by others [13,14]. To our knowledge, previous studies on the relationship between vitamin A and BPD did not control for the presence of inflammation or mechanical ventilation, in which both factors contribute to the pathogenesis of BDP [3]. In our study, first-week IL-6 levels were higher in the DEXA group compared to the No DEXA group. Moreover, all infants in the DEXA group received mechanical ventilation. Both factors were independently associated with lower retinol concentrations at one week of age. The presence of inflammation is known to affect vitamin A homeostasis as IL-6 suppresses the hepatic production of RBP4, resulting in lower retinol concentrations [17,19]. CRP, another acute-phase protein, is a feasible marker of the inflammation response. In adults, even slightly elevated CRP levels have been associated with a decline in retinol concentrations [19]. The authors concluded that the interpretation of vitamin A status was not reliable when the CRP level was >10 mg/L [19]. In the present study, the association between first-week retinol and CRP concentrations was not significant, likely due to low CRP levels in the cohort; only 10% of the assessments were above 10 mg/L.

Mechanical ventilation may injure the immature lungs through baro-volume trauma and oxygen toxicity, which both up-regulate inflammatory reactions [34,35]. Thus, mechanical ventilation may indirectly suppress hepatic RBP4 synthesis via inflammatory pathways. In our study, mechanical ventilation was associated with retinol concentrations independent of IL-6 levels. It is therefore tempting to speculate that the negative correlation between mechanical ventilation and vitamin A status was not only related to inflammatory reactions but also due to the increased utilization of vitamin A for lung repair or to combat oxidative stress. A recent meta-analysis concluded that vitamin A supplementation in very preterm infants may reduce the duration of supplemental oxygen and mechanical ventilation, but the sample size for these outcomes was small and statistical heterogeneity was considerable [15]. To the best of our knowledge, the associations between inflammation, mechanical ventilation and vitamin A status in preterm infants have not previously been studied. It is not known whether low retinol concentrations observed in the first postnatal week independently enhance the risk of BPD, or whether low retinol concentrations are a result of other contributing factors including inflammation and mechanical ventilation. However, given the importance of vitamin A in cellular growth and differentiation, a transient decrease in blood concentrations may cause functional deficiencies in vitamin-A-dependent tissues.

In this study, neonates treated with dexamethasone to facilitate extubation had a two-fold higher retinol concentration at postnatal Day 28 compared to non-treated infants. Georgieff et al. described similar observations in the late 1980s [22]. They studied 13 newborn infants with respiratory disorders and reported a significant increase in plasma retinol and RBP4 during and 14 days after dexamethasone exposure. In a study of 23 preterm infants <29 weeks gestation (Shenai et al.), elevated retinol and RBP4 concentrations were observed during dexamethasone treatment but they returned to baseline values after two weeks [21]. In consistency with both of these studies, we observed a peak response during dexamethasone exposure, which was independent of vitamin A intake.

The mechanisms behind elevated vitamin A concentrations during dexamethasone exposure are not completely understood. The simultaneous increase in both retinol and RBP4 indicates that the retinol:RBP4 complex is released from the liver during exposure. In addition, the transient mobilization of vitamin A from the lung may have contributed to the elevated blood concentrations [23]. Animal models reveal that some vitamin A is stored in the lungs during perinatal development to facilitate local growth and differentiation, as well as for the synthesis of surfactants [7,16]. Previous studies suggest that the potential maturational influence of dexamethasone in the premature lungs is partly linked to the increased pulmonary supply and utilization of vitamin A [7,21,22,23,24]. In a study of preterm infants with respiratory morbidity (Shenai et al.), each 0.35 µmol/L increment in retinol concentrations was associated with a 60% increase in the odds favoring a positive pulmonary response to dexamethasone treatment, indicated by successful weaning from supplemental oxygen and mechanical ventilation [21]. Whether the transient increase in vitamin A concentrations during dexamethasone exposure led to a favorable treatment response was not studied in our cohort of preterm infants.

Our data suggest that vitamin A intakes within the current range of recommendations (400–1000 µg/kg/d) are insufficient to provide adequate blood levels in preterm infants <29 weeks gestation. Overall, eight out of ten infants had retinol concentrations <0.7 µmol/L at 36 weeks PMA despite vitamin A intakes in line with guidelines [31]. In a randomized controlled trial (RCT) of extremely preterm infants, daily enteral doses of 3000 µg/kg/d resulted in retinol concentrations >0.7 µmol/L in less than 50% of the supplemented infants [36]. It is known that preterm infants are at risk of fat-soluble vitamin malabsorption due to low levels of pancreatic enzymes and bile salts in the intestinal lumen [37]. Retinyl palmitate (preformed vitamin A), which requires hydrolysis and bile salts to form micelles before absorption can occur [17], was the main source of vitamin A in our study. Thus, ineffective absorption may partly explain the high prevalence of vitamin A deficiency and why vitamin A intakes did not correlate with blood concentrations. Two recent RCTs demonstrated enhanced retinol concentrations after enteral provision with a high dose of a water-soluble vitamin A supplement compared to a placebo [38,39], but only one of these studies found beneficial effects of supplementation on respiratory outcomes [39].

We observed a significant negative effect of being born SGA on retinol concentrations at Day 7 after adjusting for mechanical ventilation and inflammatory markers. A subgroup analysis from a large multicenter RCT on intramuscular vitamin A injections to prevent BPD revealed a positive effect on retinol concentrations after supplementation only in extremely preterm infants born appropriate for gestational age (AGA) [40]. The number of SGA infants in our work and the cited study was low compared to the AGA subgroups, affecting the robustness of the observed associations between SGA and vitamin A status. Some SGA infants are growth-restricted at birth due to placental insufficiency, resulting in poor nutrient accretion. Whether the amount of vitamin A needed to compensate for the fetal deficits among these infants is above the needs of AGA-born peers, or whether vitamin A metabolism differs between these subgroups, remains unknown.

### Strengths and Limitations

Among the strengths of this study are the multimodal evaluation of vitamin A status, including longitudinal measurements of retinol, RBP4 and inflammatory markers, and the assessments of vitamin A intake and dexamethasone exposure. Limitations include the observational design and incomplete set of biomarkers among some participants. Vitamin A is sensitive to light degradation [41]. In our study, whole blood samples were collected on Mitra^®^ sticks and dried in room air without light protection. Thus, exposure to light might have resulted in degradation and lower vitamin A concentrations compared to actual blood values. However, retinol obtained with DBS showed a high correlation (r = 0.813) with measured plasma retinol concentrations. Analyses of vitamin A status, non-nutritional factors and clinical outcomes were not powered to study causal associations, which is why our results should be regarded as exploratory and hypothesis generating.

## 5. Conclusions

This comprehensive assessment of vitamin A status in preterm infants <29 weeks gestation showed that the majority of the infants exhibited biochemical deficiency during hospitalization, despite vitamin A intakes within the range of current recommendations. Exposure to dexamethasone seems to mask this deficiency. The optimal retinol concentration in preterm infants remains undefined, but our observations suggest that the presence of inflammation and dexamethasone exposure should be considered when vitamin A status is interpreted.

## Figures and Tables

**Figure 1 nutrients-15-00441-f001:**
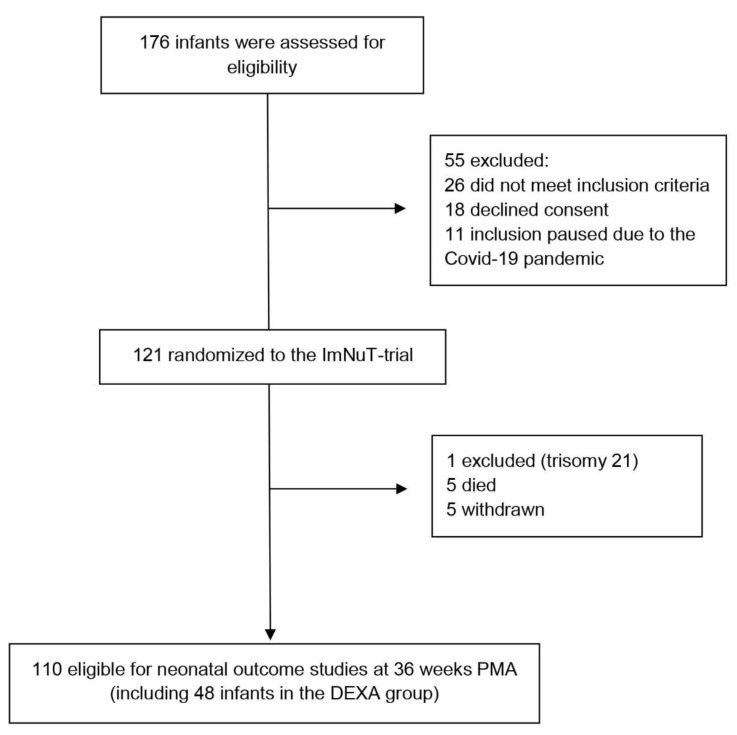
Flow diagram of study participants. PMA, postmenstrual age. DEXA group, infants who received dexamethasone treatment during hospitalization.

**Figure 2 nutrients-15-00441-f002:**
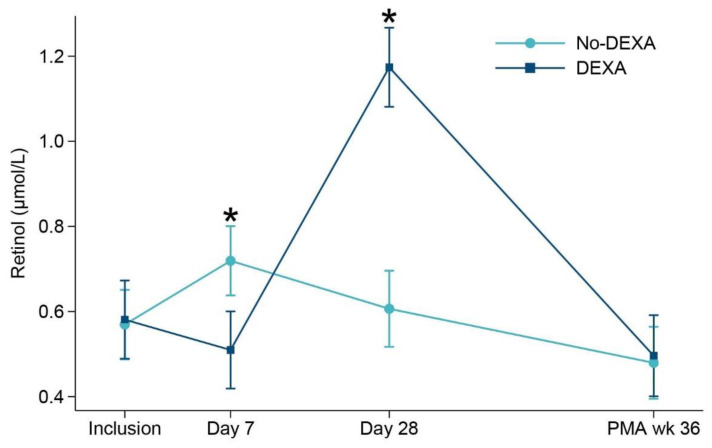
Longitudinal measures of retinol concentrations according to dexamethasone group. Day 7 (*p* = 0.001) and Day 28 (*p* < 0.001). Number of infants with vitamin A measurements at each study time point. Inclusion *n* = 99, Day 7 *n* = 96, Day 28 *n* = 83 and 36 weeks PMA *n* = 86. PMA, postmenstrual age. * *p* ≤ 0.001.

**Figure 3 nutrients-15-00441-f003:**
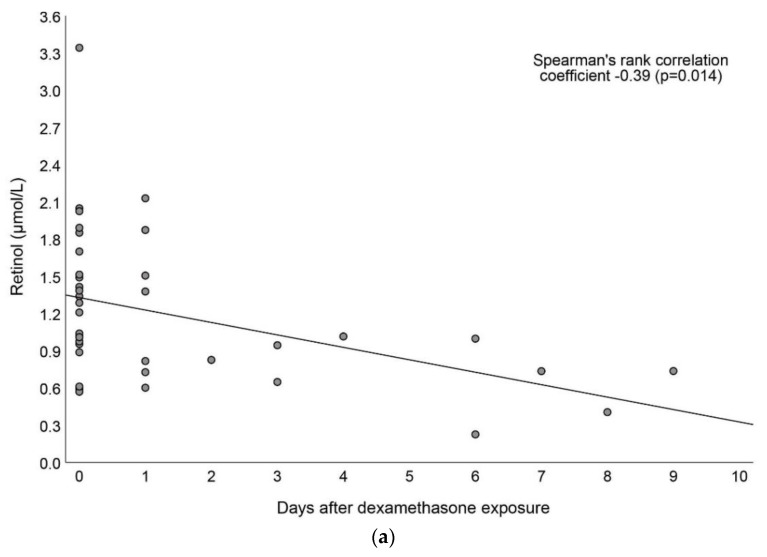
Associations between (**a**) retinol (*n* = 37) and (**b**) RBP4 (*n* = 37) and days after dexamethasone exposure. Ongoing treatment with dexamethasone was defined as 0 days. The study period included the first 14 days after treatment began. RBP4, retinol-binding protein 4.

**Figure 4 nutrients-15-00441-f004:**
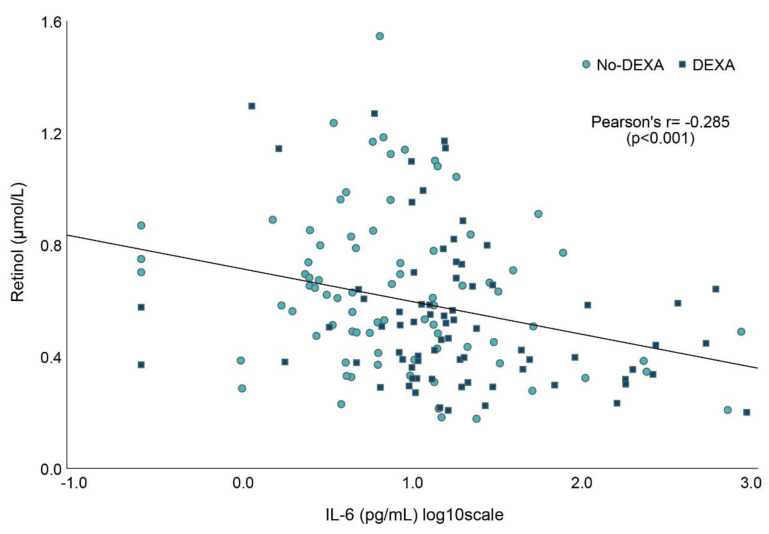
First-week associations between blood concentrations of IL-6 and retinol (*n* = 163). Paired measurements were obtained at inclusion (within 48 h of life) and at Day 7. The figure includes markers according to dexamethasone group. IL-6, Interleukin-6.

**Table 1 nutrients-15-00441-t001:** Characteristics and clinical outcomes.

	ImNuT Cohort (*n* = 120)	No DEXA (*n* = 71)	DEXA (*n* = 49)
Maternal characteristics			
Antenatal glucocorticoids any dose, *n* (%)	120 (100)	71 (100)	49 (100)
Antenatal glucocorticoids 2 doses, *n* (%)	83 (69)	53 (75)	30 (61)
Birth characteristics			
Gestational age, weeks, median (range)	26^+6^ (22^+6^–28^+6^)	27^+4^ (23^+5^–28^+6^)	24^+5^ (22^+6^–28^+6^)
Weight, g, median (IQR)	798 (666, 1070)	995 (720, 1140)	712 (602, 804)
Weight, z score, median (IQR)	−0.88 (−1.7, −0.29)	−1.1 (−2.1, −0.32)	−0.74 (−1.3, −0.23)
Female, *n* (%)	54 (45)	34 (48)	20 (41)
SGA, *n* (%)	23 (19)	17 (24)	6 (12)
Apgar score at 5 min, median (IQR)	8 (6, 8)	8 (7, 9)	7 (6, 8)
CRIB score, median (IQR)	6 (2, 9)	4 (1, 7)	9 (7, 11)
Clinical outcomes			
NEC, *n* (%)	4 (3)	3 (4)	1 (2)
Mechanical ventilation, days, median (IQR)	5 (0, 23)	0 (0, 3)	26 (17, 35)
BPD, *n* (%) ^a^	49 (41)	13 (21)	40 (83)
ROP, *n* (%) ^b^			
No ROP	70 (64)	45 (73)	25 (52)
Stage ≥ 3	10 (9)	2 (3)	8 (17)
IVH, *n* (%)			
No IVH	95 (79)	64 (90)	31 (63)
Grade 3–4	11 (9)	5 (7)	6 (12)
PDA requiring treatment, *n* (%) ^c^	51 (45)	15 (23)	36 (74)
Any septicemia, *n* (%)	61 (51)	27 (38)	34 (69)

DEXA, dexamethasone; SGA, small for gestational age; CRIB, Clinical Risk Index for Babies; NEC, necrotizing enterocolitis; BPD, bronchopulmonary dysplasia; ROP, retinopathy of prematurity; IVH, intraventricular hemorrhage; PDA, patent ductus arteriosus.

**Table 2 nutrients-15-00441-t002:** Multivariable linear regression analyses between possible predictors and retinol concentrations at Day 7.

	Retinol (log_10_ µmol/L) Day 7		
	Beta	Standard Error	*p*-Value
MV at day 7 (yes = 1)	−0.19	0.047	<0.001
SGA (yes = 1)	−0.17	0.056	0.003
log_10_ IL-6 (pg/mL)	−0.085	0.040	0.038
log_10_ CRP (mg/L)	−0.085	0.050	0.094
Adjusted R^2^	0.367		

MV, mechanical ventilation; SGA, small for gestational age; IL-6, Interleukin-6; CRP, C-reactive protein. Data are presented as unstandardized beta coefficients with their standard errors. Sex, GA, SGA status, CRIB score, mechanical ventilation at Day 7, total vitamin A intake from birth to Day 7 and blood concentrations of CRP and IL-6 (on log_10_ scale) were included as possible predictor variables. Only infants with a complete dataset were included in the analyses (*n* = 65).

## Data Availability

The data presented in this study are available on request from the corresponding author. The data are not publicly available due to GDPR restrictions and we do not have legal approval to share individual data.

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
