# Peer review of "Vitamin A Status in Preterm Infants Is Associated with Inflammation and Dexamethasone Exposure"

_nutrients, 2023, doi:10.3390/nu15020441_

Round 1
Reviewer 1 Report
Thank you for the opportunity to review this manuscript. It is an interesting article.
In the present study, authors assess vitamin A status during hospitalization in preterm infants < 29 weeks gestation. The term “Vitamin A” (VA) refers to fat-soluble retinoids, including retinol, retinal, and retinyl esters. It has many properties.
I have some suggestions to improve this manuscript. In the introduction, I think it's important to illustrate the properties of vitamin A. In particular, it's necessary to mention antiviral e antibacterial properties. Here are recent references to add "Sinopoli, A.; Caminada, S.; Isonne, C.; Santoro, M.M.; Baccolini, V. What Are the Effects of Vitamin A Oral Supplementation in the Prevention and Management of Viral Infections? A Systematic Review of Randomized Clinical Trials. Nutrients 2022, 14, 4081. https://doi.org/10.3390/nu14194081"Gürbüz, M., & Aktaç, Åž. (2022). Understanding the role of vitamin A and its precursors in the immune system. Nutrition Clinique et Métabolisme." I suggest providing the lector with other information on the mechanism of action of vitamin A during transit in the organism. Methods and results are well described. In the discussion can be useful to compare the results of this study with other study that evaluated this topic.
Author Response
Point 1: I have some suggestions to improve this manuscript. In the introduction, I think it's important to illustrate the properties of vitamin A. In particular, it's necessary to mention antiviral e antibacterial properties. Here are recent references to add "Sinopoli, A.; Caminada, S.; Isonne, C.; Santoro, M.M.; Baccolini, V. What Are the Effects of Vitamin A Oral Supplementation in the Prevention and Management of Viral Infections? A Systematic Review of Randomized Clinical Trials. Nutrients 2022, 14, 4081. https://doi.org/10.3390/nu14194081"Gürbüz, M., & Aktaç, Åž. (2022). Understanding the role of vitamin A and its precursors in the immune system. Nutrition Clinique et Métabolisme." I suggest providing the lector with other information on the mechanism of action of vitamin A during transit in the organism.
Response 1: Thank you for reviewing our manuscript and for your valuable comments and suggestions. We are happy that you found the manuscript interesting and for providing additional references. We particularly found the review article by Gürbüz and Aktac (2022) relevant to our research article. As preterm infants are at high risk of concomitant diseases associated with inflammation, we agree that it is relevant to provide details on the role of vitamin A in the immune system and its mechanisms of action. Considering you suggestions, we have added some more details and cited the article by Gürbüz and Aktaç (2022) in the introduction section (P2, L46-52).
Point 2: In the discussion can be useful to compare the results of this study with other study that evaluated this topic.
Response 2: Thank you for your comment. We comprehensively searched the literature during the conceptualization and preparation of the present research article. Although the role of vitamin A in premature morbidities have been studied for decades, we could not find studies that have assessed vitamin A status in preterm infants using a multimodal approach, combining longitudinal measurements of vitamin A biochemistry, dietary assessment, biomarkers of inflammation and clinical outcome data. To the best of our knowledge, there is a lack of evidence regarding associations between vitamin A and inflammatory processes in premature infants. In the discussion of our results, we have therefore also compared our data with findings from the pediatric and adult literature. The associations between dexamethasone exposure and vitamin A concentrations observed in our study, in addition to the possible mechanisms behind this relationship, are discussed and put in context with previous studies on the matter. To our opinion, we have tried our best to compare and discuss the present results with relevant studies. Thus, we have not made any additions to the discussion section of the revised manuscript.
Point 3 (from the questionnaire):
Question #1 Are all cited references relevant to the research?
Answer #1: Must be improved
Response 3: Considering you suggestions to relevant references for the manuscript, we have cited the article by Gürbüz and Aktaç (2022) in the introduction section (reference #5). To our opinion, we have tried our best to compare and discuss the present results with evidence from relevant studies. We have critically reviewed the relevance of the cited references during this revision process, and have not made any further change to the reference list of the revised manuscript.
Reviewer 2 Report
Authors performed an observational study to assess vitamin A status in preterm infants born before 29 weeks of gestational age, in relation to dexamethasone treatment and to markers of inflammation. The study appears, to my opinion, very well performed and descripted. Statistical analysis is very robust, and the study is very in depth discussed. I have not comments, and to my opinion the study could be accepted also in present form. I have only one curiosity before the publication. In line 84 you talk about “critical illness with short life expectancy”, how did you define this? In addition, I suggest to add the p-value of table 1 (DEXA vx No-DEXA)
Congratulations
Author Response
Point 1: I have only one curiosity before the publication. In line 84 you talk about “critical illness with short life expectancy”, how did you define this?
Response 1: Thank you for reviewing our manuscript and for your question. We are happy that you found the study well performed and described. We did not define “critical illness with short life expectancy” in the study protocol. Infants (n=15) considered to be in danger of imminent death (based on the opinion of the physician in charge) was not included in the study.
Point 2: In addition, I suggest to add the p-value of table 1 (DEXA vx No-DEXA)
Response 2: Thank you for your suggestion. The purpose of Table 1 is to provide a presentation of the characteristics and clinical outcomes of the ImNuT-cohort and between the dexamethasone groups. We do not find p-values for baseline differences between the groups relevant for this research article, since they are not testing any of the scientific hypotheses of the present study. To study associations between retinol concentrations and possible predictor variables, including characteristics and clinical outcome data, we included data from the total ImNuT-cohort in the regression analyses. For these reasons, we have chosen not to present p-values for the between-groups comparisons in Table 1.